# Microplastics in the Gulf of Mexico: A Bird's Eye View

**Jacquelyn K. Grace [1,2,*], Elena Duran [2], Mary Ann Ottinger [3], Mark S. Woodrey [4] and Terri J. Maness [5]**

[1] Department of Ecology and Conservation Biology, Texas A&M University, College Station, TX 77843, USA
[2] Ecology and Evolutionary Biology Interdisciplinary Program, Texas A&M University, College Station, TX 77843, USA; emiduran@tamu.edu
[3] Department of Biology and Biochemistry, University of Houston, Houston, TX 77204, USA; maotting@central.uh.edu
[4] Department of Wildlife, Fisheries, and Aquaculture & Coastal Research and Extension Center, Mississippi State University, Mississippi State, MS 39762, USA; msw103@msstate.edu
[5] School of Biological Sciences, Louisiana Tech University, Ruston, LA 71272, USA; tmaness@latech.edu
[*] Correspondence: jacquelyn.grace@ag.tamu.edu; Tel.: +1-979-458-9871

**Abstract:** Microplastic debris is a persistent, ubiquitous global pollutant in oceans, estuaries, and freshwater systems. Some of the highest reported concentrations of microplastics, globally, are in the Gulf of Mexico (GoM), which is home to the majority of plastic manufacturers in the United States. A comprehensive understanding of the risk microplastics pose to wildlife is critical to the development of scientifically sound mitigation and policy initiatives. In this review, we synthesize existing knowledge of microplastic debris in the Gulf of Mexico and its effects on birds and make recommendations for further research. The current state of knowledge suggests that microplastics are widespread in the marine environment, come from known sources, and have the potential to be a major ecotoxicological concern for wild birds, especially in areas of high concentration such as the GoM. However, data for GoM birds are currently lacking regarding typical microplastic ingestion rates uptake of chemicals associated with plastics by avian tissues; and physiological, behavioral, and fitness consequences of microplastic ingestion. Filling these knowledge gaps is essential to understand the hazard microplastics pose to wild birds, and to the creation of effective policy actions and widespread mitigation measures to curb this emerging threat to wildlife.

**Keywords:** plastic; coastal birds; marine birds; debris; seabirds; Gulf of Mexico; microplastic





## 1. Introduction

Plastic polymers have been in production since the mid-19th century and reached exponential growth during the 1950s [1]. Since then, global annual plastic production has increased nearly 310 times, from 1.5 million tons in 1950 to 370 million tons in 2019 [2]. Plastics are essentially non-biodegradable and easily transported by water and air currents [1], and thus, their disposal poses a persistent, global pollution problem. Disposed and discarded plastics at various stages of breakdown are contributing to the growing quantities of litter on beaches and coastal areas, particularly on continents [3]. In 2010, approximately 4.8–12.7 million tons of plastic were discharged annually from the land into the ocean with inputs projected to increase 10-fold by 2025 [4,5]. In 2013, the global estimate of plastic in the oceans was 5.25 trillion plastic pieces weighing 268,940 tons [6] and ranging in size from nanometers to meters. Despite much attention on environmental plastics in recent years, their actual amounts, distribution, and ecological significance is still unclear. This is largely due to the recency of attention, lack of adequate systematic sampling and analysis, and the immensity and diversity of the problem [7].

While all plastic debris can have negative effects on organisms, in this review we focus on microplastics and ultra-fine plastics (Table 1) because they are produced in large quantities in the Gulf of Mexico region, are small enough to be ingested by a wide range of organisms, and a relatively large volume of published literature has been produced

on their abundance and effects as compared to the relatively newer research areas of nanoplastics. Microplastics and smaller plastic particles are also of great environmental concern compared to larger plastics because their small size and variable distribution in marine waters and sediments make monitoring and collection for removal/recycling difficult [8,9]. We refer to both micro- and ultra-fine plastics as "microplastics" here, because these two categories are rarely distinguished in the existing literature that we review [10]. However, for future studies we recommend the categories suggested by Provencher et al. ([11]; Table 1) as technological advances allow for discrimination between smaller size categories.

**Table 1.** Categorizations of plastic debris by size (data used to construct this table from [10]). It should be noted that these categorizations are not universal, and much confusion exists regarding the upper and lower size limits for microplastics.

| Size Range | Category |
|---|---|
| 1 nm–1 µm | Nanoplastic |
| 1 µm–1 mm | Ultra-fine plastic |
| 1 mm–5 mm | Microplastic |
| 5 mm–20 mm | Mesoplastic |
| 20 mm–100 mm | Macroplastic |
| >100 mm | Megaplastic |

Microplastics are either purposefully produced (i.e., primary microplastics) or produced as a by-product of the degradation of larger macroplastics (i.e., secondary microplastics) [10,12], through environmental degradation, or organismal digestion processes [13]. Primary microplastics are predominantly microbeads that are produced as precursors to the manufacture of larger plastics (i.e., nurdles), or for personal care products, cleaning agents, oil and gas drilling fluids in the oil and gas industry, and coatings and paints [8]. Microplastic pollution on beaches was first noted in the 1970's and has since garnered much scientific attention [4]. Today, microplastics are a ubiquitous global pollutant in oceans, estuaries, freshwater, and even arctic ice [8]. Nurdles are of particular concern because of the large scale of their manufacture, particularly along the Gulf Coast states of Texas and Louisiana (Figure 1), and because by their nature they are hard to contain [14]. Beginning in the 1940's, nurdles have been produced and shipped around the world, where they are melted down and turned into larger plastic products [4,5]. Nurdles can thus be lost at the manufacturing facility, in transport, loading, storage, and/or at the fabrication destination (Figure 1; [14]). Plastic fragments (e.g., irregular pieces degraded from larger plastics) and fibers are other major contributors to microplastic pollution, along with plastic film, ropes, filaments, sponges, foams, rubber, and microbeads in decreasing order of importance [10,12].

In this review, we focus on the known and predicted effects of microplastics on birds in the Gulf of Mexico (GoM) and identify knowledge gaps that should be filled to enable scientifically sound mitigation. Microplastics pose a potential emerging global challenge for the health, resilience, and sustainability of coastal bird populations. Birds are useful bioindicators of ecosystem health because they are typically high in the food chain, ubiquitous, and relatively easily captured and sampled [15]. Birds can thus be sentinel species of environmental change with implications for other wildlife taxa, humans, and domestic species. Here, we review existing scientific peer-reviewed and gray literature (e.g., theses, dissertations, governmental, and non-governmental reports) to provide a brief overview of the state of knowledge of microplastic pollution in the GoM, followed by a summary of known effects of microplastics and associated chemicals on birds, and methodological considerations for field and laboratory studies, including priority bird species in the GoM for field studies. We conclude with identifying major gaps in knowledge and recommendations, both in the laboratory and the field, that are needed to understand the extent and severity of the hazard posed by microplastic pollution to avian wildlife.

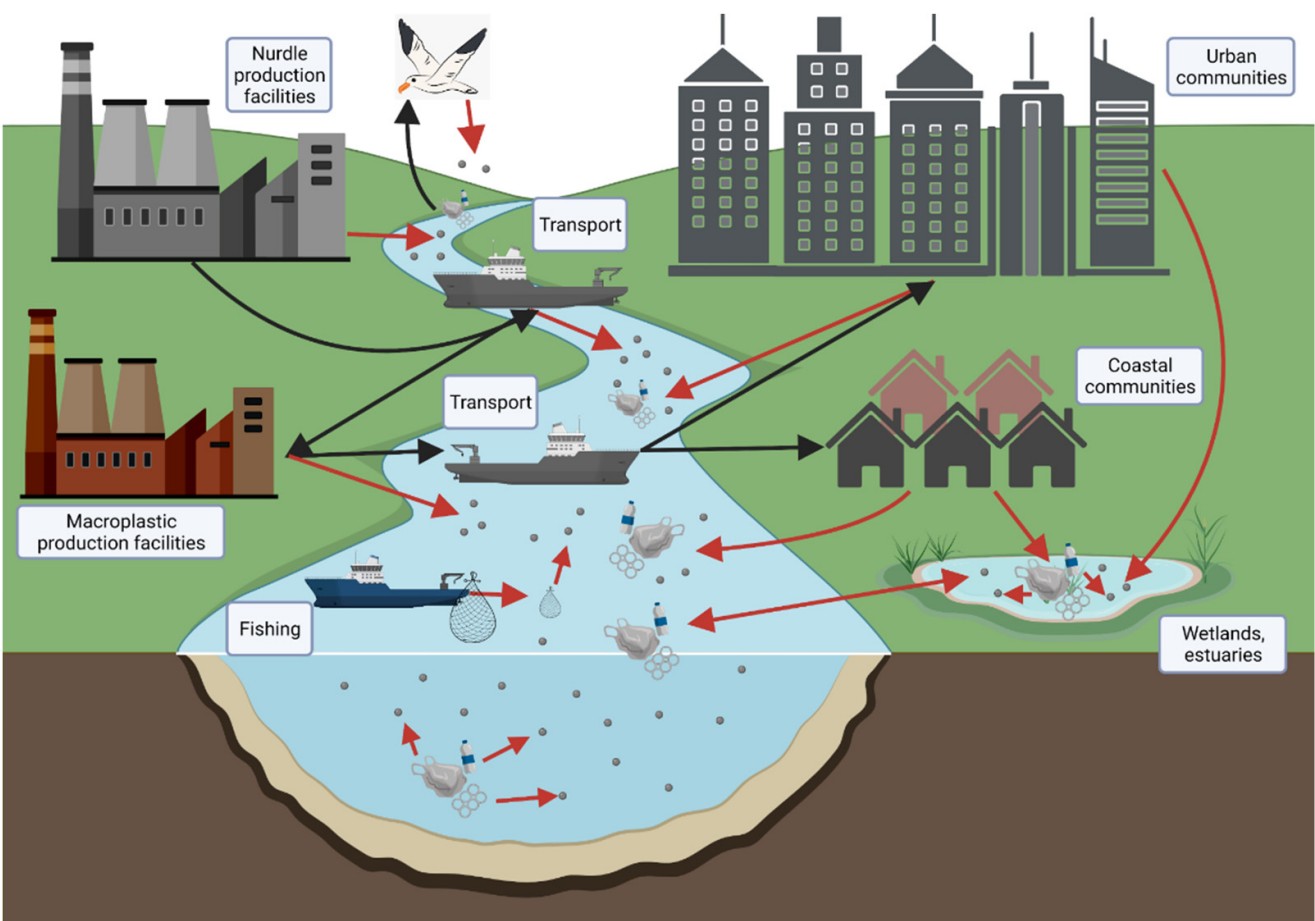

**Figure 1.** Pathways by which microplastics enter the marine environment. Black arrows indicate movement of plastics across the landscape; red arrows indicate disposal of plastics in waterways. Small gray spheres represent microplastics, which can enter marine waterways directly as primary microplastics or indirectly through breakdown of macroplastics. (Created with BioRender.com, accessed on 28 May 2022).

## 2. Microplastics in the Gulf of Mexico

The Gulf of Mexico (GoM) is the largest gulf in the world, with a depth of 4384 m, surface area of 615,000 square miles, and width of 810 miles. It connects the Atlantic Ocean and Caribbean Sea and is bound in the north by the United States, in the south/southwest by Mexico, and in the southeast by Cuba [12]. The GoM is of special interest for microplastic pollution because it is home to most plastic manufacturers in the United States, predominantly in and around Galveston Bay, Texas, as well as elsewhere on the Texas and Louisiana coasts (Figure 2; [4]). Moreover, the GoM is subject to additional plastic discharges from the continents, port areas, tourism activities (which can have both positive [3] and negative effects [16,17] on beach cleanliness), river systems, and industrial activities [12]. Globally, semi-enclosed marine bodies of water appear to be areas of high microplastic accumulation, and unsurprisingly, concentrations of microplastics reported off the coast of Louisiana, in the northern GoM, are among the highest reported, worldwide [18].

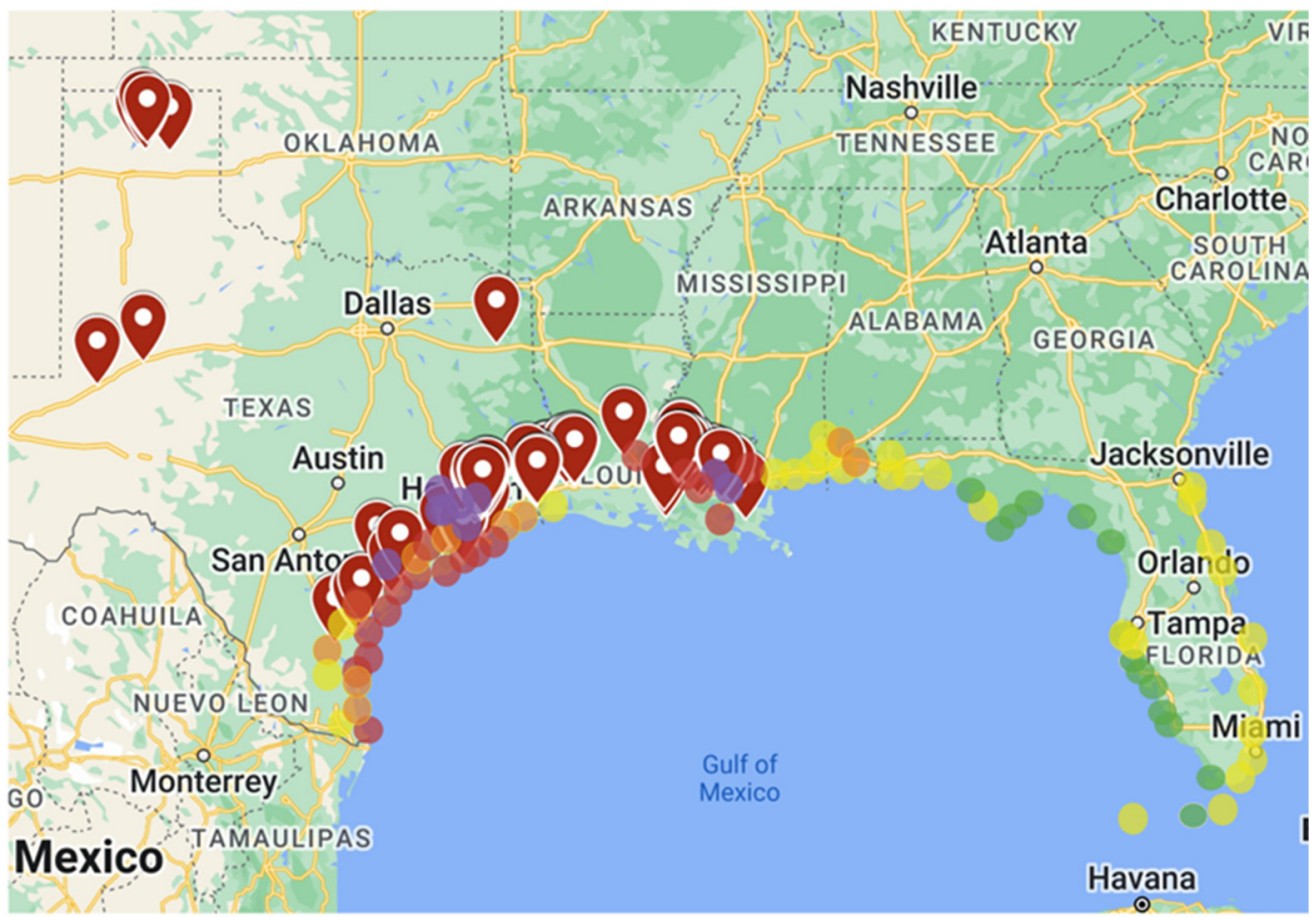

**Figure 2.** Map of the northern Gulf of Mexico showing generalized locations of plastic manufacturing facilities and nurdles collected along shorelines between November 2018 and May 2022 by Nurdle Patrol surveys. Plastic manufacturing facilities are indicated by red pins, and the approximate number of nurdles collected are represented by colored circles. Green circles indicate approximately 0 nurdles/10 min, yellow indicates 1–30 nurdles/10 min, orange indicates 31–100 nurdles/10 min, red indicates 101–1000 nurdles/10 min, and and purple indicates >1000 nurdles/10 min. Plastic facility location data is from BeyondPlastics.org [19], and nurdle data is approximated from NurdlePatrol.org [20]. For exact nurdle collection data the reader is referred to Nurdle Patrol.org.

One of the greatest ecotoxicological concerns regarding microplastics is their ability to adsorb other chemicals [21]. The Gulf of Mexico is an area of high population density, agricultural, military, and industrial activity all of which contribute to chemical contamination of waters and shorelines. Over 45% of the total U.S. petroleum refining capacity, and 51% of the total U.S. natural gas processing plant capacity is located along the Gulf Coast. Additionally, offshore production in the GoM accounts for 17% of total U.S. crude oil production and 5% of dry natural gas production [22]. The high concentrations of toxins associated with oil production or discharged from agricultural and industrial activities may adsorb to microplastics, increasing their toxicity for wildlife.

### 2.1. Distribution of Microplastics in the Gulf of Mexico

Microplastics can accumulate in sediment, the water column, and in biota. In the GoM sediment, plastics generally fall into the ultrafine to microplastic size categories (0.2–5 mm), and are typically composed of polypropylene (PP), polyethylene (PE), polystyrene (PS), polyethylene terephthalate (PET), polyester, polyamide (PA), or nylon microplastics. Most sedimentary ultrafine/microplastics in the GoM are hard plastics (23%), followed by

fragments (18%), rigid or semi-rigid (14%), fibers (10%), foam (8%), film (6%), strand (5%), and others (16%), and colors including blue, white, transparent, red, black, green, yellow [12]. Microplastics appear to be widespread, regardless of area conservation status. For example, Weitzel et al. [23] examined sediment samples from wildlife management areas and conservation preserves and found microplastics in 64% of samples.

With regard to distribution, there appears to be a gradual decrease in sedimentary microplastic concentration from west to east in the northern GoM, as determined from sand samples collected at loggerhead turtle nesting sites [24]. Similarly, a Gulf-wide assessment of beach nurdles in 2018–2019 found the highest standardized counts in Texas, with the highest site being Galveston Bay (30,846 nurdles collected in 10 min) [4]. Across the GoM, the highest concentrations of sedimentary nurdles coincided with nurdle production facilities [4], suggesting that these bays are sources, not sinks, for microplastics. Although sediment sampling is the most widespread form of microplastic monitoring, current knowledge is limited both by depth of sediment and habitats sampled. Most sediment sampling for plastics, to date, have been at the sediment surface (i.e., 0–5 cm in depth; [12]). Sampling of sediments occurs almost exclusively along beaches and rarely in marshes and other less accessible habitats [23]. Since small plastic particles can be filtered and retained at a variety of sedimentary depths, patterns of plastic distribution in sediments across the GoM remain uncertain.

In the GoM water column, microplastics range in size from 10–1730 μm, and are largely composed of irregular fragments (75%), fibers (21%), and a small percentage of beads (3.5%) [12]. Microplastic types in the GoM water column include PE, alkyd resin, cellophane, polyester, polyurethane (PU), acrylate, PA, PP, polyacetal, polytetrafluoroethylene, PET, and polyvinyl chloride (PVC) [12]. Concentrations and geographic distribution of microplastics in the GoM water column can be difficult to estimate, because of strong tidal variation [17]. However, a 1992 United States Environmental Protection Agency study of 14 harbors along the coastal United States found the highest concentration of nurdles in the Houston Ship Channel, with 700,344 pellets, followed by New York/New Jersey harbor with 11,266 pellets [25]. Thus, the concentration of microplastics in the GoM water column, and especially in and around Houston, Texas, appears to be quite large compared to the rest of the continental United States.

Geographically, microplastic concentration varies with proximity to marine waters and/or microplastic sources and with tidal patterns [26]. At least one study suggests that bays have higher exposure to microplastic than both estuaries and the open ocean [26]. For example, concentrations of microplastics in Mobile Bay, Alabama, USA were 66–253 times higher than those of the open ocean [26]. Leeward bays with strong tidal influence appear to be particularly vulnerable to microplastic accumulation [26]. However, this pattern may be altered by proximity to microplastic sources. In Mobile Bay, microplastics appear to accumulate from marine deposition; however, other locations such as the Houston Ship Channel may be sources of microplastics due to the large numbers of nurdle producing facilities, which may alter deposition patterns. Similarly, proximity to large rivers and urban areas significantly increase microplastic abundance along the southeastern United States coast, probably because wastewater discharge is a major source of microplastics [27]. However, there is still much we do not know about microplastic abundance and distribution in the Gulf of Mexico. For instance, no studies to date have evaluated marine subsurface microplastics, below 10 m in water depth, or sediment microplastics below the top 0–5 cm [12]. In addition, the lack of consensus on units for reporting microplastic abundance makes it difficult to compare between studies [11,12]. Finally, GoM-wide distribution studies are also rare, and given the strong influence of tides and seasonality, additional studies are needed to gain a full picture of factors influencing microplastic distribution in the GoM and to predict areas of high risk.

Work is currently underway to address these areas of needed research in the GoM through the Gulf of Mexico Alliance (GOMA), for which marine debris is a top priority. GOMA's marine debris cross-team initiative (MDCTI) [28] published suggested

guidelines [29] for monitoring microplastics in water and sediments. MDCTI focus areas include research (i.e., to understand where marine debris originates and increase monitoring), prevention (i.e., to raise awareness and improve stewardship), and removal (i.e., to locate, remove, and dispose of debris). The group supports projects in each of these areas with a focus on community science by using the EPA's Escaped Trash Assessment Protocol [30], the Marine Debris Tracker App [31], the Nurdle Patrol App [20], and NOAA's Marine Debris Monitoring and Assessment Project tool [32]. Most of these community science projects and protocols (except for the Nurdle Patrol) focus on large debris and not on microplastics *per se*, but these data may provide insight into the origins of microplastics and suggest ways to prevent and mitigate microplastic pollution within the GoM.

### 2.2. Microplastics in GoM Wildlife

Most studies of microplastics in the biota of the GoM have focused on marine and freshwater fish from Texas, the Mississippi Gulf, and Veracruz. In these studies, most of the species sampled were in trophic levels 3–4, and composed of 48% demersal, 27% benthopelagic, 10% pelagic-neritic, and 5% pelagic species. Although the percentage of fish samples with microplastics varied greatly by species and sampling location, approximately 46% of GoM fish appear susceptible to microplastics. Density of plastics ingested varied between demersal and pelagic species, suggesting that microplastic density may affect dispersion in the water column [12]. All biota studies in the GoM have focused on microplastics in the intestinal tract, but smaller ultrafine-/nanoplastics can translocate from the gut to the circulatory system, then to other organs, and this has yet to be evaluated in biota from the Gulf of Mexico [12]. From existing studies of microplastics in GoM biota, it appears that most microplastics ingested are fibers (60%), followed by fragments (21%), film (11%), and finally beads or pellets (8%), and are composed of a variety of plastic types (e.g., PE, PP, PVC, PET, nylon, acrylic, epoxy resin; [12]).

To date, few studies have evaluated microplastics in GoM birds. The species studied thus far (Table 2) include clapper rails (*Rallus crepitans*) and seaside sparrows (*Ammospiza maritima*) in the Mississippi Gulf Coast tidal marshes [23]; birds of prey in Central Florida [33]; domestic chickens (*Gallus gallus*) in traditional Mayan home gardens in Pucnachen, Campeche, Mexico [34]; and brown pelicans (*Pelecanus occidentalis*), royal terns (*Thalasseus maximus*), laughing gulls (*Leucophaeus atricilla*), and double-crested cormorants (*Phalacrocorax auritus*) in South Florida [35]. Nearly all clapper rails, seaside sparrows, seabirds, and all birds of prey examined contained microplastics (Table 2). In chickens, data suggests that plastics are broken into smaller pieces as they travel through the digestive tract, going from macroplastic particles at ingestion (all plastics in crops were >5 mm; $11.0 \pm 15.3$ particles/crop) to ultrafine particles at excretion (plastics in feces ranged from 0.1–1 mm; $129.8 \pm 82.3$ particles/g; [35]). However, quantities of microplastics per bird varied widely in these studies and were generally low when stomach flushing was used on live birds, as compared to dissection of the gastrointestinal contents of dead birds (Table 2). Stomach flushing may not recover as many microplastic particles as lethal sampling and dissection [23]. Type of microplastic also varied by species, with most of the particles found in chickens from PE-bottle debris (91.4%), followed by fibers (6.9%) [34], while fibers were the leading type of microplastic in seabirds (72% of microplastics) followed by plastic fragments (28%) [36]. Among seabirds, brown pelicans had the most microplastics ($29.9 \pm 20.1$ particles/bird) and laughing gulls had the least ($7.6 \pm 4.6$ particles/bird; [35]).

**Table 2.** Microplastic ingestion studies in Gulf of Mexico birds showing species, sampling location, sample size (N), sampling method, percentage of birds sampled with microplastics in their gastrointestinal tracts, the number of microplastic particles found (SD = standard deviation, IQR = interquartile range), plastic type, and study citation. * Numerical data presented here were recalculated from raw data provided with the cited published study (see Source column).

| Species | Location | N | Sampling Method | % with Micro-Plastic | Number of Microplastic Particles | Plastic Type | Source |
|---|---|---|---|---|---|---|---|
| Domestic chicken (*Gallus gallus*) | Pucnachen, Campeche, Mexico | 50 | Necropsy | Not given | None (all were macroplastic) median = 11.0 (SD = 15.3) particles/crop | 91.4% Polyethylene bottle debris, 6.9% fibers, and 1.7% polystyrene | [34] |
| Domestic chicken (*Gallus gallus*) | Pucnachen, Campeche, Mexico | 50 | Necropsy | Not given | Microplastic: median = 10.2 (SD = 13.8) particles/gizzard; macroplastic: median = 45.8 (SD = 2.6) particles/gizzard | 91.4% Polyethylene bottle debris, 6.9% fibers, and 1.7% polystyrene | [34] |
| Domestic chicken (*Gallus gallus*) | Pucnachen, Campeche, Mexico | 20 | Feces (10 g/ea) | Not given | None (all were ultrafine) median = 129.8 (SD = 82.3) particles/g | Not given | [34] |
| Clapper Rail (*Rallus crepitans*) | Mississippi coastal marsh | 35 | Lavage | 83% | Median = 6.0 (SD = 7.2) particles/stomach (micro & ultrafine particles combined) | 99% fibers | [23] |
| Seaside Sparrows (*Ammospiza maritima*) | Mississippi coastal marsh | 36 | Lavage | 69% | Median = 2.0 (SD = 2.7) particles/stomach (micro & ultrafine particles combined) | 98% fibers | [23] |
| Red-shouldered Hawk (*Buteo lineatus*) | Central Florida | 28 | Necropsy | 100% | Median = 19.5 (IQR = 32.2) Mean = 24.8 (SD = 23.4) | 37% processed cellulose, 16% polyethylene terephthalate, & 11% polymer blend | [33] * |
| Osprey (*Pandion haliaetus*) | Central Florida | 16 | Necropsy | 94% | Median = 7.5 (IQR = 29.8) Mean = 20.1 (SD = 24.4) | 37% processed cellulose, 16% polyethylene terephthalate, & 11% polymer blend | [33] * |
| Barred Owl (*Strix varia*) | Central Florida | 8 | Necropsy | 100% | Median = 4.0 (IQR = 4.5) Mean = 5.2 (SD = 5.6) | 37% processed cellulose, 16% polyethylene terephthalate, & 11% polymer blend | [33] * |
| Eastern Screech Owl (*Megascops asio*) | Central Florida | 4 | Necropsy | 100% | Median = 10.5 (IQR = 5.5) Mean = 10.5 (SD = 7.8) | 37% processed cellulose, 16% polyethylene terephthalate, & 11% polymer blend | [33] * |

**Table 2.** *Cont.*

| Species | Location | N | Sampling Method | % with Micro-Plastic | Number of Microplastic Particles | Plastic Type | Source |
|---|---|---|---|---|---|---|---|
| Black Vulture (*Coragyps atratus*) | Central Florida | 2 | Necropsy | 100% | Median = 10.0 (IQR = 2) Mean = 10.0 (SD = 2.8) | 37% processed cellulose, 16% polyethylene terephthalate, & 11% polymer blend | [33] * |
| Turkey Vulture (*Cathartes aura*) | Central Florida | 2 | Necropsy | 100% | Median = 18.0 (IQR = 10) Mean = 18.0 (SD = 14.1) | 37% processed cellulose, 16% polyethylene terephthalate, & 11% polymer blend | [33] * |
| Red-tailed Hawk (*Buteo jamaicensis*) | Central Florida | 2 | Necropsy | 100% | Median = 2.5 (IQR = 0.5) Mean = 2.5 (SD = 0.7) | 37% processed cellulose, 16% polyethylene terephthalate, & 11% polymer blend | [33] * |
| Cooper's Hawk (*Accipiter cooperii*) | Central Florida | 1 | Necropsy | 100% | 9.0 | 37% processed cellulose, 16% polyethylene terephthalate, & 11% polymer blend | [33] * |
| Brown Pelican (*Pelecanus occidentalis*) | Southern Florida | 13 | Necropsy | 100% | Median = 27.0 (IQR = 24.5) Mean = 39.9 (SD = 19.5) | 77.2% fibers & 22.8% fragments | [35] * |
| Royal Tern (*Thalasseus maximus*) | Southern Florida | 9 | Necropsy | 100% | Median = 8.0 (IQR = 6.0) Mean = 9.9 (SD = 4.5) | 64.0% fibers & 36.0% fragments | [35] * |
| Laughing Gull (*Leucophaeus atricilla*) | Southern Florida | 11 | Necropsy | 100% | Median = 8.0 (IQR = 3.5) Mean = 9.0 (SD = 4.3) | 78.8% fibers & 21.2% fragments | [35] * |
| Double-crested Cormorant (*Phalacrocorax auratus*) | Southern Florida | 10 | Necropsy | 90% | Median = 9.0 (IQR = 5.2) Mean = 10.0 (SD = 8.1) | 53.1% fibers & 46.9% fragments | [35] * |

## 3. Effects of Microplastic Ingestion on Birds

Plastic ingestion has been documented in over 180 species, including birds [36]. Birds may ingest microplastics purposefully by mistaking them for edible food items [37], by trophic transfer (indirect ingestion; e.g., [34,38,39]), or incidentally through parental feeding of young (e.g., [40]), feeding modality (e.g., filter feeding [41]), or daily activities (e.g., preening feathers [41]). Here we review effects of microplastic consumption on birds and highlight areas of research that need further investigation.

### 3.1. Physical Effects of Microplastic Ingestion on Birds

Ingestion of plastics by birds can be problematic due to their largely non-digestible nature. Macroplastics are well known to cause physical blockages, damage, or a false sense of satiety in macrobiota [42]. However, it is unclear if microplastics cause similar direct physical harm to vertebrate wildlife, because their small size may allow them to pass through the gastrointestinal tract. Microplastic size, organism size, feeding modality, and gastrointestinal tract structure undoubtedly contribute to the physical impacts of ingested microplastics (e.g., [43]). Future research in this area is needed to evaluate potential physical and mechanical impacts of microplastics on birds.

### 3.2. Toxicological Effects of Microplastic Ingestion on Birds

Besides physical damage and obstruction, microplastics pose an ecotoxicological concern for wildlife because they contain chemicals associated with plastic production and can readily adsorb chemicals from the environment at later time points (Figure 3; Table 3). These chemicals can leach from plastics following ingestion by wild birds [44]. For example, micro- and mesoplastics collected from the Pacific leached several endocrine disrupting compounds (EDCs), particularly estrogenic EDCs and especially bisphenol A (BPA) [45]. Importantly, smaller microplastics leached greater quantities of EDCs than did larger plastics, because of more efficient sorption from surrounding seawater [45].

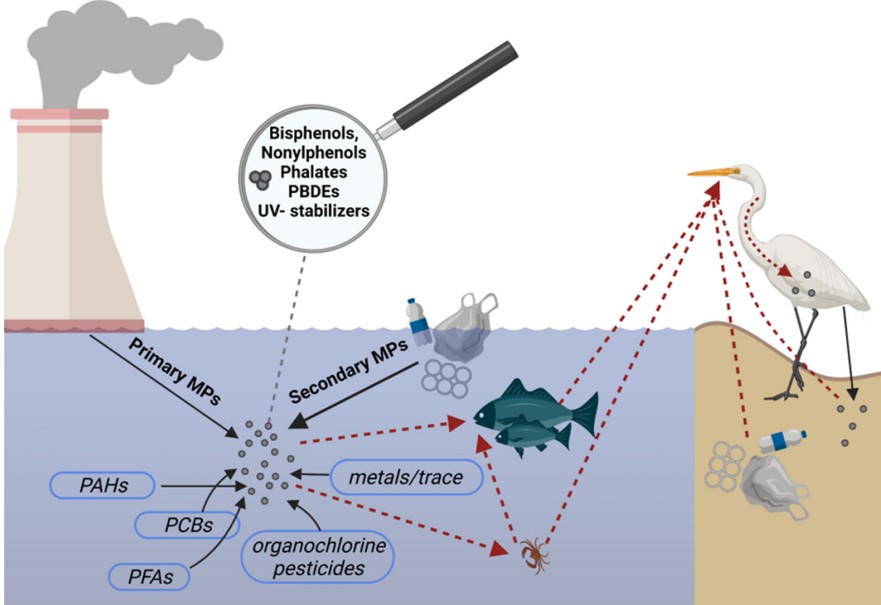

**Figure 3.** Pathways of ingestion by wildlife of microplastics (MPs) and associated chemicals. Additives are part of the plastic production process (illustrated in the magnifying glass) and additional chemicals can adsorb to plastics following production through anthropogenic use and/or environmental deposition. Black arrows indicate pathways of microplastic production and chemical adsorption. Dotted red arrows indicate pathways of wildlife ingestion. Microplastics can be directly consumed from the environment, or macroplastics that are consumed can be broken down into microplastics through digestion, partially excreted (although some probably remain within the organism), and re-ingested as microplastics (Created with BioRender.com, accessed on 28 May 2022).

**Table 3.** Toxins associated with microplastics with additional literature sources for further information. Plasticizers and plastic components are added to plastics during production, while "adsorbed by plastics" indicates that the chemical(s) is/are adsorbed from the environment after production.

| Toxin | Association to MPs | Additional Resources |
|---|---|---|
| Bisphenols & nonylphenols | Plasticizer | [21,46–54] |
| Phthalates | Plasticizer | [49,55–71] |
| PBDEs | Plastic component | [72–85] |
| UV stabilizers | Plastic component | [86–90] |
| Metals/trace metals | Adsorbed by plastics/plastic component | [66,69,89,91–113] |
| PCBs | Adsorbed by plastics | [114–124] |
| Organochlorine pesticides | Adsorbed by plastics | [117,125–129] |
| PAHs | Adsorbed by plastics | [55,129–143] |

In invertebrate and fish species, microplastic ingestion is associated with altered biomarkers of oxidative stress and genotoxic, neurotoxic, and inflammatory effects [144]. However, few studies have examined toxicological impacts of microplastic ingestion in birds beyond correlations with body mass. In the wild, establishing toxicological effects is naturally difficult, due to many confounding variables and methodological choices. For example, ingested plastics positively correlated with circulating uric acid, amylase, and cholesterol levels, and negatively correlated with blood calcium levels in fledgling flesh-footed shearwaters (*Ardenna carneipes*) [145]. However, these relationships differed depending on the quantification method for plastics (e.g., presence/absence, number of plastic pieces, plastic mass). Additionally, established serum chemistry reference intervals are often non-existent for wild birds [146], making it difficult or impossible to determine whether chemistries are outside of a normal range in relation to any potential toxin, including microplastics.

Controlled microplastic feeding experiments can alleviate issues of confounding variables; however, given the recency of attention on microplastics, very few feeding experiments have been conducted in birds, thus far. The first plastic feeding experiment was conducted in newly hatched Japanese quail (*Coturnix japonica*), using polypropylene nurdles exposed to the Tasmanian sea, to mimic environmental weathering and toxin adsorption [147]. Plastic ingestion was correlated with delayed growth (resolved at 6 weeks of age), delayed reproductive maturity in females (resolved by 7 weeks of age), and increased presence and severity of epididymal intra-epithelial cysts in male parental and offspring generations [147]. The latter two effects may be due to endocrine disruption by chemicals associated with plastics, although acute toxic effects were not observed and most measured parameters showed no evidence of endocrine disruption [147]. Microplastic ingestion has also been investigated in adult female Japanese quail, which revealed increases in biomarkers of oxidative stress (i.e., increased malondialdehyde and reactive oxygen species in liver tissue, and reduced superoxide dismutase activity) in the plastic-fed group. Additionally, systemic effects were observed, with birds who ingested microplastics exhibiting a 43% increase in malondialdehyde production in the brain [13]. However, further controlled studies examining the relationship between microplastics and endocrine disruption are necessary.

3.2.1. Effects on Birds of Individual Chemical Additives in Plastics

The effects of individual chemicals associated with microplastics on birds have been examined much more thoroughly than effects of microplastics themselves, in the laboratory. These chemicals include plasticizers (e.g., phthalates, bisphenols, and nonylphenols) and flame retardants (e.g., polybrominated diphenyl ethers, ultraviolet stabilizers) added to plastics during production, and chemicals that adsorb to plastics from the environment (Table 3). Phthalates, now environmentally ubiquitous, can come from several sources, but are most associated with plastics as a plasticizer. In birds, phthalates appear to be deposited in eggs, as well as positively correlated with concentrations of malondialdehyde in the egg,

which may increase embryonic oxidative stress [148]. Phthalates may impart several other toxic effects; for example, in salmon (*Salvelinus malma*), phthalates influence expression of immune related genes [149]. Similarly, bisphenols and nonylphenols, particularly BPA, have endocrine disrupting effects on vertebrates, and on birds specifically [150–152]. Bisphenols and nonylphenols are detectable in many aquatic environments, but modeling indicates that ingestion of microplastics is not a relevant exposure pathway for at least one marine vertebrate, the Atlantic cod (*Gadus morhua*) in the North Sea [51]. Similar research of avian species is needed to: (1) evaluate rates of leaching and uptake in vitro and in vivo for both phthalates and bisphenols/nonylphenols, (2) investigate toxicity at ecologically relevant concentrations, (3) investigate synergistic effects with other adsorbed contaminants (e.g., metals [153]), and (4) evaluate microplastics as a possible exposure pathway to these chemicals.

Polybrominated diphenyl ethers (PBDEs) are flame retardants that are applied to plastics during manufacture and are toxic to avian species, with various effects (see [154]). The evidence is quite strong that at least some PBDEs can leach from ingested microplastics and be taken up by surrounding tissues in birds. For example, in short-tailed shearwaters (*Ardenna tenuirostris*), the same highly brominated PBDE congeners (e.g., decabromodiphenyl ether, BDE 209) that are applied to plastics were detected both in the plastic fragments found in the bird's gut and in the same bird's adipose tissue [155]. Even stronger evidence comes from a feeding experiment in which polyethylene microplastics, prepared with BDE 209 and ultraviolet (UV) stabilizers, were fed to free-living, wild streaked shearwater chicks (*Calonectris leucomelas*). Fifteen to sixteen days after plastic feeding, 47% of the BDE 209 had leached from the plastics, and concentrations of BDE 209 and UV stabilizers in adipose fat and livers of exposed birds were elevated 91–120,000 times the rate from the natural diet [156]. Thirty-two days after plastic feeding, concentrations of BDE 209 and the UV stabilizers had decreased in the liver, but not adipose fat, probably due to metabolism or redistribution to other organs [156]. Specific effects of UV stabilizers on birds requires further study.

### 3.2.2. Effects on Birds of Individual Environmental Chemicals Associated with Plastics

Microplastics also pose an ecotoxicological threat to birds due to their ability to adsorb environmental chemicals. The most widespread contaminants of concern in the Gulf of Mexico include polycyclic aromatic hydrocarbons (PAHs) from oil and gas activities, metals (e.g., lead, mercury, arsenic, cadmium, silver, nickel, tine, chromium, zinc, and copper), dioxins, flame retardants, pesticides, and polychlorinated biphenyls (PCBs) [157]. Microplastics may provide an additional exposure pathway for these contaminants in birds (see [158]). Metals, PCBs, and organochlorine pesticides have received particular scientific attention regarding their potential associations with plastic debris and adverse effects on birds, while PAHs and dioxins remain understudied. PAHs, which are toxic and can act as EDCs and carcinogens (e.g., [159]), are considered persistent organic pollutants. A large body of literature demonstrates that associations between PAHs and microplastics are possible and occur in the marine environment (see [160]; Table 3), but little is known about the extent to which PAHs can leach from microplastics or be taken up by bird tissues. Dioxins, which are developmentally and reproductively toxic (e.g., [161]), can be adsorbed by microplastics (e.g., [78,162]; Table 3), but similar to PAHs, little is known about the implications of this adsorption in terms of additional exposure risk to birds.

Heavy and trace metals have long been the subject of toxicology studies in birds, and some metals (e.g., lead and mercury) have clear toxic effects on many organisms including birds (e.g., [163,164]). Trace and heavy metals can be incorporated into microplastics during manufacture [165], or they can be adsorbed by microplastics in the environment [113,166–168] (Table 3). Worryingly, microplastics can highly concentrate the metals they attract, although adsorption rates differ depending on plastic age, temperature, pH, contact time, ionic strength, and particle size [169]. Correlational data suggest that metals associated with plastics can be taken up by avian tissues [170,171], although direct experimental evidence

is lacking. Moreover, evidence from a marine fish (marine medka, *Oryzias melastigma*) suggests that uptake of metals into tissues may be facilitated by the presence of microplastics [110]. Simulations of avian digestion likewise suggest that metals can be rapidly mobilized from polyethylene microplastics [172]. According to these simulations, adverse effects would occur only with a high level of microplastic ingestion [172]; however, birds are exposed to metals in myriad other ways throughout their lifetime, and metals from microplastics must be considered in the context of these other routes of exposure. Moreover, evidence for synergistic adverse effects of BPA and metal pollutants in zebra fish (*Danio rerio*) [153] suggests that the combined adsorption of these contaminants to plastics may rapidly elevate their toxicity.

PCBs are also well documented toxins (e.g., [173]) that have been detected in plastics, generally (e.g., [78], Table 3), and in microplastics ingested by birds, specifically [174]. Highly chlorinated PCB congeners are of special concern, because these congeners are the most abundant species detected in adipose and liver tissues of bird species examined [175,176]. However, the relationships between microplastics and PCBs appear to be complex. Evidence suggests that microplastics may transfer PCBs to organisms, but also that ingested microplastics may actually adsorb dietary PCBs (see [50]). Additionally, experimental studies in other organisms (e.g., Norway lobster, *Nephrops norvegicus*) reveal minimal uptake and accumulation of PCBs from PCB-enriched plastics [177], and ingested plastics have not been correlated with tissue concentrations of PCBs in birds [175,176]. Future in vitro and in vivo studies on avian uptake of PCBs from microplastics are necessary to evaluate microplastics as potential vectors for PCB exposure.

A strong body of literature indicates that organochlorine pesticides (OCPs) are also able to bind to plastics and have been detected in environmental microplastics [178] (Table 3). OCPs include such well-known chemicals as dichlorodiphenyltrichloroethane (DDT), which was brought to public attention in Rachel Carson's "Silent Spring" [179] and has since received significant scientific attention regarding its toxic effects on birds (e.g., [180]). *p,p'*-dichlorodiphenyldichloroethylene (*p,p'*-DDE), the breakdown product of *p,p'*-DDT, was the highest concentrated OCP detected in plastic pellets and fragments from the ventriculus and proventriculus of eight *Procellariiformes* species on the east coast of Brazil [174]. As with most other chemicals associated with microplastics, however, we currently lack evidence regarding OCP uptake by avian tissues from microplastics, and interactions between plastic type and OCP uptake.

In summary, a significant body of research demonstrating the capability of microplastics to adsorb environmental contaminants exists (Table 3), but to date, little evidence suggests that microplastics can act as vehicles for exposure to toxic levels of these contaminants in birds. Three important aspects to consider when assessing the risk of exposure of birds to environmental chemicals through plastic ingestion are: (1) the potential for the chemical to adsorb to microplastics, (2) the potential for leaching from the microplastic and subsequent uptake of the chemical upon ingestion/digestion, and (3) the potential for exposure to lead to adverse effects [181]. In consideration of these three conditions, current evidence is sparse for significant exposure leading to adverse effects in organisms from chemicals adsorbed by microplastics [181]. This point has been further substantiated, for example, in northern fulmars (*Fulmarus glacialis*), for which concentrations of POPs in tissues are not correlated with concentrations of POPs in ingested plastics [182]. Further study is extremely important, however, as the amount of leaching, uptake, and thresholds for adverse effects likely differs by chemical, plastic type, species, and life stage of exposed birds.

## 4. Methodological Recommendations for Microplastic Research in Birds

Our review of the literature has identified significant gaps in knowledge that must be filled to enable scientifically sound mitigation strategies and policy changes for microplastics in the Gulf of Mexico (Figure 4). These gaps will be best addressed through both field monitoring and experiments and laboratory experiments. Below, we present

methodological recommendations for both field and laboratory research in this area and highlight additional literature sources in which readers can find more specific methodological protocols.

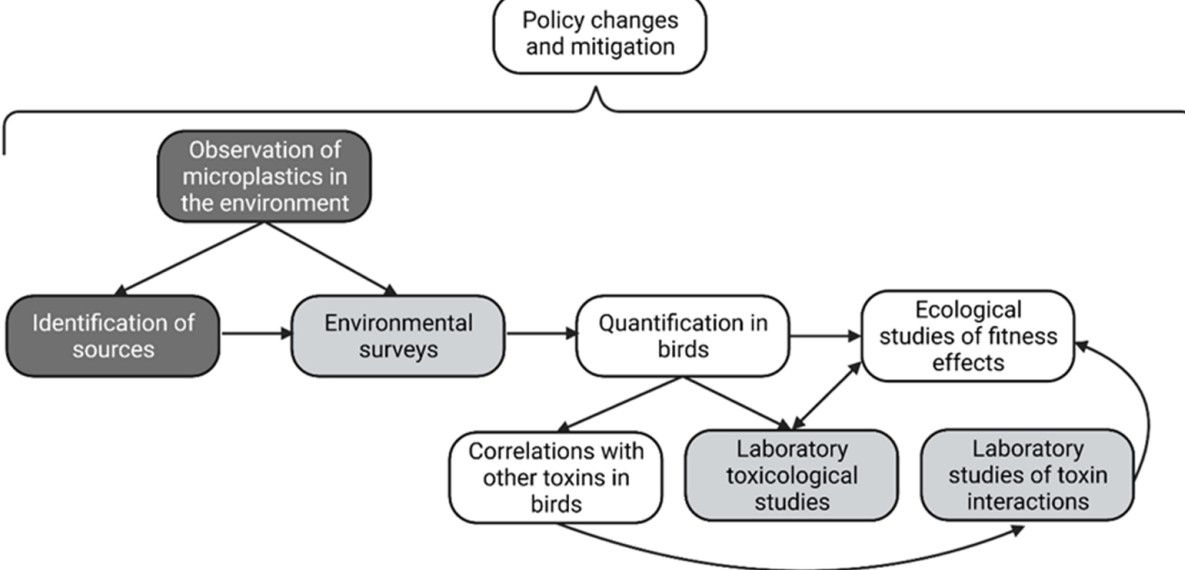

**Figure 4.** Scientific pathway to create sound policy and techniques to mitigate the effects of environmental microplastics on birds. Dark gray boxes are steps that are largely completed, light gray boxes are steps that are partially completed, and white boxes are steps that are largely incomplete for birds in the Gulf of Mexico. (Created with BioRender.com, accessed on 28 May 2022).

### 4.1. Field Sampling Design, Collection, and Processing

Many field studies of plastic ingestion by birds are opportunistic with samples collected from necropsies of birds found dead, in bycatch, or confiscated from illegal hunting [10], which can bias population trends. More systematic sampling is needed to assess and monitor population trends in plastic ingestion. In this section, we discuss considerations for selecting individuals to sample, how to collect and process samples, and relevant data to be recorded. Importantly, we recommend that researchers adopt standardized nomenclature for particle sizes (Table 1) to avoid confusion and to facilitate comparisons across studies.

#### 4.1.1. Sample Design

Given the lack of microplastic studies involving birds across the Gulf of Mexico region, we recommend a two-tiered framework of avian species for regional and local monitoring of microplastics (Table 4). Tier I species include common and wide-spread species that allow for robust evaluation of microplastic exposure and ingestion across the entirety of the geography of the GoM and across-taxa comparisons while also allowing flexibility of local-scale studies specific to a particular objective (see "Quantification in birds" node, Figure 4). Tier II includes those which have a limited distribution, are only present in the GoM seasonally, or are species of conservation concern where the role of microplastics could potentially be a mechanism behind their elevated conservation status. For consistency with current avian monitoring efforts across the Gulf, we follow the Gulf of Mexico Avian Monitoring Network's Birds of Conservation Concern list for Tier II species [183]. For research-focused studies, i.e., those that would fall under the "Ecological studies of fitness effects" node in our conceptual Scientific Pathways model (Figure 4), we suggest researchers focus on Tier I species due to their common and widespread status as well as the relative ease of securing permits to capture, sample (especially targeted lethal sampling), or hold them in captivity for laboratory studies.

**Table 4.** Recommended bird species for monitoring and understanding the impacts of microplastics across the Gulf of Mexico region. Taxa are based on Wilson et al. [183], feeding guild on de Graaf et al. [184], and landcover/resident status on species-specific Birds of the World accounts [185]. For explanation of Tier designation see "Section 4.1.1 Sample design".

| Tier | Species | Taxa | Feeding Guild | Landcover Association | Resident status in GoM | Game Species |
|------|---------|------|---------------|-----------------------|------------------------|--------------|
| I | Northern shoveler (*Spatula clypeata*) | Waterfowl | Omnivore: freshwater strainer | Saltmarshes, estuaries, lakes, flooded fields, wetlands, agricultural ponds, and wastewater ponds | Wintering | Yes |
| I | Blue-winged teal (*Spatula discors*) | Waterfowl | Omnivore: freshwater dabbler | Fresh and brackish marshes | Year-round, wintering | Yes |
| I | Red-breasted merganser (*Mergus serrator*) | Waterfowl | Piscivore: coastal diver | Coastal estuaries and coastal bays | Wintering | Yes |
| II | Mottled duck (*Anas fulvigula*) | Waterfowl | Omnivore: marsh dabbler | Fresh and brackish marshes | Year-round | Yes |
| II | Lesser scaup (*Aythya affinis*) | Waterfowl | Omnivore: water bottom forager | Ponds, lakes, reservoirs, coastal estuaries, and coastal bays | Wintering | Yes |
| II | Northern pintail (*Anas acuta*) | Waterfowl | Omnivore: coastal beach forager, coastal dabbler | Lakes, reservoirs, estuaries, saltmarshes, freshwater and brackish wetlands, and bays | Wintering | Yes |
| I | Clapper Rail (*Rallus crepitans*) | Marsh bird | Omnivore: Saltmarsh forager (nonbreeding); crustaceovore, molluscovore: saltmarsh gleaner (breeding) | Brackish and saltwater marshes | Year-round | Yes |
| I | Sora (*Porzana carolina*) | Marsh bird | Omnivore: marsh forager | Fresh, brackish, and salt marshes | Wintering | Yes |
| II | King rail (*Rallus elegans*) | Marsh bird | Crustaceovore: marsh prober (breeding) & Omnivore: saltmarsh prober (nonbreeding) | Fresh and brackish water marshes | Year-round | Yes |
| II | Seaside sparrow (*Ammospiza maritima*) | Marsh bird | Crustaceovore, molluscovore, insectivore saltmarsh gleaner | Brackish and saltwater marshes | Year-round, wintering | No |
| II | Yellow Rail (*Coturnicops noveboracensis*) | Marsh bird | Omnivore: marsh forager | Fresh, brackish, and saltwater marshes; wet line savanna | Wintering | No |
| II | American bittern (*Botaurus lentiginosus*) | Marsh bird | Carnivore: water ambusher | Freshwater marshes | Wintering | No |

**Table 4.** *Cont.*

| Tier | Species | Taxa | Feeding Guild | Landcover Association | Resident status in GoM | Game Species |
|---|---|---|---|---|---|---|
| II | Least bittern (*Ixobrychus exilis*) | Marsh bird | Piscivore, insectivore: water ambusher | Fresh and brackish water marshes | Breeding, year-round | No |
| II | Nelson's sparrow (*Ammospiza nelsoni*) | Marsh bird | Omnivore: ground forager | Brackish and saltwater marshes | Wintering | No |
| I | Great egret (*Ardea alba*) | Wading bird | Carnivore: water ambusher | Freshwater, brackish, and marine wetlands | Year-round | No |
| I | Green heron (*Butorides virescens*) | Wading bird | Carnivore: water ambusher | Marine and freshwater wetlands | Year-round | No |
| II | Snowy egret (*Egretta thula*) | Wading bird | Carnivore: water ambusher | Estuaries, saltmarshes, tidal channels, shallow bays, and mangroves | Year-round | No |
| II | Sandhill Crane (*Antigone canadensis*) | Wading bird | Omnivore: fresh- marsh forager, ground forager | Grasslands, wetlands | Wintering | Yes |
| II | Wood stork (*Mycteria americana*) | Wading bird | Carnivore: water ambusher | Fresh and brackish water marshes | Year-round/wintering | No |
| II | Little blue heron (*Egretta caerulea*) | Wading bird | Carnivore: water ambusher | Swamps, marshes, ponds, streams, lagoons, tidal flats, canals, ditches | Year-round, wintering | No |
| II | Tricolored heron (*Egretta tricolor*) | Wading bird | Carnivore: water ambusher | Coastal estuaries, salmarshes, mangroves, and lagoons | Year-round, wintering | No |
| II | Sedge wren (*Cistothorus stellaris*) | Marsh bird | Insectivore: ground gleaner | Grasslands & marshes | Wintering | No |
| II | Marsh wren (*Cistothorus palustris*) | Marsh bird | Insectivore: marsh gleaner | Wetlands, tidal saltmarshes, and weedy agricultural canals | Wintering | No |
| I | Black-bellied plover (*Pluvialis squatarola*) | Shorebird | Carnivore: coastal beach gleaner | Mudflats and beaches | Wintering | No |
| I | Killdeer (*Charadrius vociferus*) | Shorebird | Insectivore: shoreline/ground gleaner | Sandbars, mudflats, and grazed fields | Year-round | No |

**Table 4.** *Cont.*

| Tier | Species | Taxa | Feeding Guild | Landcover Association | Resident status in GoM | Game Species |
|---|---|---|---|---|---|---|
| I | Willet (*Tringa semipalmata*) | Shorebird | Carnivore: coastal beach prober | Beaches, bay shores, marshes, mudflats, and rocky coastal zones | Wintering | No |
| I | Sanderling (*Calidris alba*) | Shorebird | Carnivore: coastal beach prober & freshwater shoreline gleaner | Sandy beaches, tidal mudflats, rocky coastlines | Wintering | No |
| II | Wilson's plover (*Charadrius wilsonia*) | Shorebird | Carnivore: coastal beach gleaner | Salt flats and sandy beaches | Breeding | No |
| II | Snowy plover (*Charadrius nivosus*) | Shorebird | Carnivore: shoreline gleaner | Sandy shorelines | Year-round, wintering | No |
| II | Dunlin (*Calidris alpina*) | Shorebird | Carnivore: coastal beach gleaner/prober | Estuaries and coastal lagoons | Wintering | No |
| II | Buff-breasted sandpiper (*Calidris subruficollis*) | Shorebird | Insectivore: ground gleaner | Grasslands | Migration | No |
| II | Western sandpiper (*Calidris mauri*) | Shorebird | Carnivore: coastal beach gleaner/prober | Fresh and saltwater marshes | Wintering | No |
| II | Long-billed curlew (*Numenius americanus*) | Shorebird | Omnivore: shoreline forager/gleaner/prober | Wetlands, tidal estuaries, mudflats, flooded fields, and beaches | Wintering | No |
| II | Marbled godwit (*Limosa fedoa*) | Shorebird | Carnivore: coastal beach prober | Coastal mudflats, estuaries, and sandy beaches | Wintering | No |
| I | Laughing gull (*Leucophaeus atricilla*) | Seabird | Carnivore: coastal beach scavenger; piscivore: coastal surface gleaner, coastal food pirate | Bays, estuaries, and landfills | Year-round, wintering | No |
| I | Double-crested cormorant (*Nannopterum auritum*) | Seabird | Piscivore: water diver | Lakes, lagoons, ponds | Year-round, wintering | No |
| I | Brown pelican (*Pelecanus occidentalis*) | Seabird | Piscivore: coastal plunger | Estuarine, coastal, nearshore | Year-round | No |
| II | Black skimmer (*Rynchops niger*) | Seabird | Piscivore: water skimmer | Shorelines | Year-round, wintering | No |
| II | Royal tern (*Thalasseus maximus*) | Seabird | Piscivore: coastal plunger | Shorelines | Year-round, wintering | No |

**Table 4.** *Cont.*

| Tier | Species | Taxa | Feeding Guild | Landcover Association | Resident status in GoM | Game Species |
|---|---|---|---|---|---|---|
| II | Sandwich tern (*Thalasseus sandvicensis*) | Seabird | Piscivore: coastal plunger | Shorelines | Year-round, wintering | No |
| II | Common loon (*Gavia immer*) | Seabird | Piscivore: coastal diver | Shorelines | Wintering | No |
| II | Audubon's shearwater (*Puffinus lherminieri*) | Seabird | Piscivore: pelagic diver | Islands | Breeding | No |
| II | Magnificent frigatebird (*Fregata magnificens*) | Seabird | Piscivore: coastal surface gleaner, coastal food pirate | Coasts and islands | Wintering | No |
| II | Northern gannet (*Morus bassanus*) | Seabird | Piscivore: coastal plunger | Oceans | Wintering | No |
| II | Osprey (*Pandion haliaetus*) | Raptor | Piscivore: water foot-plunger | Rivers, lakes, reservoirs, lagoons, swamps, and marshes | Year-round, wintering | No |
| II | Bald eagle (*Haliaeetus leucocephalus*) | Raptor | Piscivore: water foot-plunger | Forested areas near water | Year-round, wintering | No |
| I | Grackle spp. (*Quiscalus* spp.) | Landbird | Omnivore: ground forager | Fresh, brackish, and saltwater marshes | Year-round, wintering | No |
| I | Common nighthawk (*Chordeiles minor*) | Landbird | Insectivore: air screener | Coastal sand dunes and beaches, grasslands | Breeding, migration | No |
| I | Red-winged blackbird (*Agelaius phoeniceus*) | Landbird | Omnivore: ground forager/gleaner | Fresh, brackish and saltwater marshes | Year-round | No |
| I | Tree swallow (*Tachycineta bicolor*) | Landbird | Insectivore: air screener | Fields, marshes, shorelines, wooded swamps | Wintering | No |

### 4.1.2. Sample Collection

Necropsies provide the opportunity to sample different segments of the gastrointestinal tract to determine if plastics change in size as they move through the alimentary tract (e.g., [34]). Necropsies are more likely to reveal the presence of microplastics since non-lethal sampling such as induced regurgitation (i.e., lavage or emetic) may not work well in species that do not regurgitate regularly [186,187] and are more likely to leave particles behind [188]. Other nonlethal sampling methods such as collection of pellets or feces may be difficult to assign to particular species, individuals, sexes, or ages. Some effort has been made to evaluate additional non-lethal methods of estimating plastic burdens, including quantification of plastic additives in preen oil, but this is unreliable and not currently recommended [186]. Accordingly, systematic lethal sampling may be preferred over non-lethal methods, except for species of conservation concern. Common and widespread species may be selected for targeted lethal sampling or sampling of game species can be coordinated with local hunters (Table 4). However, lethal sampling precludes the possibility of repeated sampling of the same individual over time.

### 4.1.3. Sample Processing

Provencher et al. [186] provide an excellent review for processing samples derived from different sources (e.g., necropsy, regurgitation). Care must be taken to minimize sample contamination, such as rinsing all instruments and containers before and after each step with filtered distilled water, wearing cotton lab coats and clothing, and keeping the sample covered. Controls, including all processing steps, should be done with no biological material present to assess potential contamination. We recommend using stacked sieves of 5 and 1 mm to ensure that isolated plastics are microplastics. Meso- and macroplastics will not pass through the 5 mm sieve and plastic particles smaller than 1 mm will not be retained. Additional smaller sieves can be used to collect ultrafine particles if they are of interest. Particles that are smaller than 1 mm likely pass through the gastrointestinal tract [186] and do not accumulate in the gastrointestinal tracts of birds.

Metrics that should be reported have been reviewed in Provencher et al. [11]. Briefly, researchers should separate plastics by category (i.e., primary or secondary) and type (i.e., fragment, fiber, pellet, film, or foam). This information can be used to identify potential sources of the plastic and suggest mitigation measures. Polymer type may give insight into associated contaminants and can be identified using Raman spectrometry or Fourier-transform infrared spectroscopy [186]. In addition, studies should report the location, timing and method of sampling, the number of birds examined, and the proportion of individuals examined with plastics in their gastrointestinal tracts. Most studies report the mean ($\pm$SD or 95% CI) number of plastic particles found [11], but the median, range, and interquartile range may be more appropriate for positively skewed distributions. Standardized reporting metrics will facilitate comparisons across species or within the same species in different regions. Studies should report where plastics were found within the gastrointestinal tract. It should be noted that plastics can be broken into smaller pieces or worn down as they pass through the gastrointestinal tract [34,189], so the number of plastic particles may not indicate the number of plastic particles ingested. Instead, the total mass of particles found may be a better indication of the amount ingested. Thus, the mean mass ($\pm$SD or 95% CI) should be reported as well. Finally, the color of the plastics ingested may indicate color preferences of particular species. In addition, plastic color can influence the structure of the microbial community associated with that plastic [190] and thus the gut microbiome potentially influencing behavior through the brain-gut-microbiota axis [191]. Other information to note includes the age, sex, and morphological measurements of the individual sampled. If lethally sampled, then tissues can be preserved for analysis of contaminants and/or physiological condition according to established protocols. Once standard protocols are adopted and more systematic sampling and monitoring implemented, we will have a better idea of population trends in plastic ingestion by birds across

the GoM. This will be an important step in understanding the potential negative effects of microplastics on GoM birds.

### 4.2. Laboratory Experimental Designs

To date, most field studies of microplastics have been observational, noting the presence of microplastics in the water column, sediment, and biota along coastal and marsh regions, and occasionally correlating these counts with body mass or condition of birds. These studies, which may be primarily monitoring in nature, provide valuable insights that correlate the presence of microplastics with effects observed in wildlife [192]. However, they are limited by a variety of confounding variables (e.g., reproductive activities, foraging and diet effects, weather effects, compounding anthropogenic effects) and limitations (e.g., timing and route of exposure, age of organism). Further studies in the laboratory in which birds are dosed with known amounts of microplastics are beginning to characterize the physiological effects of ingestion and associated chemical breakdown products, although much more work is necessary.

The United States Environmental Protection Agency (EPA) has guidelines for toxicological studies in birds, both for short-term catastrophic high exposures that may be lethal and longer sublethal exposures [193]. There are several different feeding experimental protocols that can be considered for testing plastics and plastic-associated chemicals, including extended One Generation testing and the Multigeneration Test. Each of these testing paradigms have measurement endpoints [193]. In the case of extended One Generation testing endpoints include lethality, orienting behavior and vigor; maturation; sexual behavior; reproductive performance and timing; weights and appearance of organs; hormones (stress, thyroid, and steroids); and molecular and cellular indicators of toxicity, health, and immune function. Multigenerational testing evaluates these endpoints in both the parental and offspring generations to determine relative vulnerability at different life stages. This also allows separating out stage-specific effects, notably developmental abnormalities due to maternal deposition of chemicals that expose the embryo during sensitive developmental stages.

## 5. Conclusions and Future Directions

### 5.1. Policy and Mitigation

In reviewing the available literature, the growing global environmental contamination from microplastics is clear. However, the potential hazard from microplastics to wild birds remains uncertain. More research on the effects of microplastics on wildlife will be essential to the development of scientifically sound mitigation and policy strategies focused on coastal birds (Figure 4). However, given the exponential growth in microplastic pollution over the past decades, we urge policy makers and industry leaders to reduce plastic waste as rapidly as possible. Macroplastic pollution is a well-documented physical hazard to wildlife [42] and has negative economic impacts due to disruption of fishing and tourism activities [194]. Tourism appears to provide a strong incentive for coastal waste management (e.g., [3]) and increased emphasis on tourism and wildlife activities may convince policymakers and local parties to implement clean-up and disposal activities. Because much microplastic pollution is produced through degradation of macroplastics [10], reduction of macroplastic debris would simultaneously reduce microplastic debris.

Several initiatives focused on reducing plastic use and pollution are currently underway in the northern GoM region. As mentioned previously, goals of GOMA's MDCTI include the prevention and removal of marine debris. To that end, MDCTI has funded education outreach programs focused on marine debris and microplastics (e.g., [195,196]) and initiatives to promote the use of plastic alternatives (e.g., [197]). Additionally, the Florida Sea Grant program's Florida Microplastic Awareness Project focuses on educating the public about the problems of marine debris [198], and Plastic Free Florida promotes citizen action for policy change to limit single-use plastics [199]. Finally, the National Oceanic and Atmospheric Administration's Marine Debris Program works with partners

in all Gulf States to promote removal efforts, develop and implement prevention strategies and action plans, and execute education and outreach. Most mitigation efforts to date have focused on education, outreach, and removal of macro- and microplastics because laws limiting single-use plastic consumption and disposal are unlikely to succeed in the current political atmosphere of the northern GoM. Three of the five GoM states (Texas, Florida, and Mississippi) have state laws in place that preempt any local plastic control ordinances, essentially prohibiting local ordinances that regulate plastic products, especially auxiliary containers [200–202].

*5.2. Current Gaps in Knowledge*

The GoM provides critical habitat for both resident and migratory bird species [183–203]. In the case of residential species and depending on their foraging areas, there is the potential for consistent exposure to microplastics. Migratory birds may be intermittently exposed to microplastics if they feed in an area replete in these contaminants while they refuel for their further journey either north or south. Currently, large gaps exist in our knowledge of the distribution of microplastics in habitats, accumulation in biota, and vulnerable species in the Gulf of Mexico. Further research is needed to evaluate the potential hazard that microplastics and associated chemicals pose for individual birds and collectively for the health and fitness of populations.

More broadly, additional research is needed to evaluate physical effects of microplastic ingestion (e.g., blockage, abrasion), rates of uptake by avian tissues of chemicals associated with plastics, and the risks of toxicological effects in birds. As mentioned above, the US Environmental Protection Agency has developed a range of guidelines for determining the potential deleterious effects of exposure to a range of contaminants. Many of these tests focus on frank toxicity and less on non-lethal effects of chemicals. However, the potential for microplastics to release their component chemicals also raises the possibility of non-lethal effects that may impair physiological processes and affect survival and/or reproduction in the wild (e.g., interfering with oxygen transfer, energy related processes, or endocrine function). Future studies should evaluate such potential sub-lethal effects in the laboratory and the field.

*5.3. Recommendations for Future Studies*

Below, we have compiled a brief list of field and laboratory studies that we recommend to advance our understanding of the hazard microplastic debris poses to wild birds in the GoM and beyond:

(1) Targeted long-term monitoring of plastics categorized by appropriate size classes (Table 1) in the sediment, water column, and biota, especially in combination with metadata on weather patterns, flooding, inputs from freshwater rivers, and proximity to plastic sources ("Environmental surveys", Figure 4). Concentrations of microplastics can be strongly influenced by season, tidal patterns [12,26]; thus, long-term monitoring is needed for a comprehensive understanding of microplastic distribution in the GoM. These data would allow for model development to overlay areas of intensive rookeries, breeding sites, and wintering foraging areas with the presence of hazardous levels of microplastics.

(2) Additional studies of microplastic presence and concentrations in wild birds in the GoM ("Quantification in birds", Figure 4). Current information is too sparse to draw conclusions regarding vulnerable species, average microplastic burdens, or risk factors for microplastic exposure. We recommend that researchers focus on Tier 1 species (Table 2) where possible. Lethal studies will be most useful to achieve this aim, but non-lethal methods (e.g., lavage, emetics) should be used where lethal studies are not possible or recommended (e.g., with species of conservation concern).

(3) Correlations between quantities and types of microplastics ingested by wild birds in the GoM and chemicals associated with plastics in tissues and feathers ("Correlations with other toxins in birds", Figure 4). Again, we recommend that researchers focus on

Tier 1 species (Table 2). Studies should also consider the potential for biomagnification of microplastics and associated chemicals, especially in high trophic level species such as raptors.

(4) Evaluation of avian tissue uptake of chemicals through microplastic ingestion ("Laboratory toxicological studies", Figure 4). Although direct uptake of additives from ingested microplastics is clear for some additives of concern (e.g., PBDEs, metals), it has not been evaluated thoroughly for most chemicals associated with plastics (e.g., PAHs, phthalates and bisphenols/nonylphenols, PCBs, OCPs). PFAs are additional emerging contaminants of concern and should also be evaluated in association with microplastics [204,205]. This should be accomplished through controlled studies feeding microplastics with known initial chemical concentrations to birds and measuring uptake in various tissues, and will most likely involve laboratory studies, but field studies will also be beneficial for ecological relevance and for species not amenable to laboratory conditions (e.g., [156]).

(5) Assessment of microplastic ingestion as a relevant pathway to sub-lethal toxic effects in avian species ("Laboratory toxicological studies", Figure 4). Controlled studies feeding microplastics with known initial chemical concentrations to birds and measuring physiological and behavioral endpoints to elucidate possible toxic effects, or lack thereof, will accomplish this objective. Studies should evaluate both single chemicals and ecologically relevant chemical cocktails ("Laboratory studies of toxin interactions", Figure 4). In most cases these studies will be conducted in the laboratory, but field studies can also achieve this aim within appropriate systems (i.e., access to nestlings, individual identification, philopatry, or individual tracking).

(6) Investigations of fitness effects of microplastic ingestion in birds ("Ecological studies of fitness effects", Figure 4). Accomplishment of this objective will involve both laboratory and field studies. For laboratory studies, controlled feeding experiments of ecologically relevant concentrations of microplastics will reveal effects on reproduction and longevity in captive conditions. Field studies will be necessary to provide correlational and experimental data (e.g., through wild feeding or microplastic removal studies) to substantiate captive findings. Correlational data may include repeated measurements of individuals over time in conjunction with longevity and reproductive data and/or measurements of microplastic loads immediately after reproductive activity in conjunction with reproduction success measures.

(7) Development of biomarkers and evaluation of biomarkers of microplastic and associated chemical toxicity in birds will be critical to assess the extent of hazard from microplastics ("Laboratory toxicological studies", "Correlations with other toxins", "Ecological studies of fitness effects", Figure 4). This development will require an understanding of the constituent chemicals and their actions. This may include microarrays, metabolomics, and individual metrics that are indicative of physiological effects.

**Author Contributions:** Conceptualization: J.K.G., M.A.O., M.S.W. and T.J.M.; review and interpretation of the relevant literature: all authors; writing—original draft preparation: J.K.G., E.D., M.A.O. and T.J.M.; writing—Review & Editing: all authors; visualizations: J.K.G., E.D. and T.J.M.; tables: T.J.M., E.D., M.S.W. and M.A.O.; project administration: J.K.G. and T.J.M.; supervision: J.K.G. and T.J.M. All authors have read and agreed to the published version of the manuscript.

**Funding:** ED was supported by a Texas A&M University Merit Fellowship. This publication is a contribution of the Mississippi Agricultural and Forestry Experiment Station. This material is based upon work that is supported by the National Institute of Food and Agriculture, U.S. Department of Agriculture, Hatch project under accession number 7002261. Any opinions, findings, conclusions, or recommendations expressed in this publication are those of the author(s) and do not necessarily reflect the view of the U.S. Department of Agriculture.

**Institutional Review Board Statement:** Not applicable.

**Informed Consent Statement:** Not applicable.

**Data Availability Statement:** Not applicable.

**Acknowledgments:** We would like to thank the anonymous reviewers of this manuscript for their helpful editorial comments and corrections.

**Conflicts of Interest:** The authors declare no conflict of interest. The funders had no role in the design of the study; in the collection, analyses, or interpretation of data; in the writing of the manuscript, or in the decision to publish the results.

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
