# Peer review of "Microplastics in the Gulf of Mexico: A Bird’s Eye View"

_sustainability, doi:10.3390/su14137849_

Round 1
Reviewer 1 Report
This review is a good quality article with a lot of information supported most of the time by sufficient bibliography. We know plastic pollution is clearly a global problem and it is certainly not exclusive to aquatic organism. Moreover, this paper focuses on birds, which often do not get sufficient attention from an ecological point of view in the context of plastic pollution. Nevertheless, there are corrections to be made:
Lines 60-63: A proper citation is missing here.
Line 214: The scientific names of species must always be written in italics. Please standardise the entire text accordingly.
Lines 264-265 and 371: In this review, the authors refer to various pollutants including bisphenol and also metals that can be transported by plastics into organisms and spread through the trophic chain. In this regard, I recommend looking at this paper DOI:10.3390/toxics9120344 in which the synergistic toxicity of bisphenol A and heavy metals was demonstrated.
Author Response
We thank the reviewer for their comments on our manuscript and provide responses to each point, below.
Lines 60-63: A proper citation is missing here.
- We have added this citation
Line 214: The scientific names of species must always be written in italics. Please standardise the entire text accordingly.
- We apologize for the formatting error and have corrected this.
Lines 264-265 and 371: In this review, the authors refer to various pollutants including bisphenol and also metals that can be transported by plastics into organisms and spread through the trophic chain. In this regard, I recommend looking at this paper DOI:10.3390/toxics9120344 in which the synergistic toxicity of bisphenol A and heavy metals was demonstrated.
- We thank the reviewer for this suggestion and have added this citation to both the bisphenol section and the metals section:
- Bisphenols (lines 338-339)- "Similar research of avian species is needed to: (1) evaluate rates of leaching and uptake in vitro and in vivo for both phthalates and bisphenols/nonylphenols, (2) investigate toxicity at ecologically relevant concentrations,(3) investigate synergistic effects with other adsorbed contaminants (e.g., metals [56]), and (4) evaluate microplastics as a possible exposure pathway to these chemicals."
- Metals (lines 390-392)- "Moreover, evidence for synergistic adverse effects of BPA and metal pollutants in zebra fish (Danio rerio) [56] suggests that the combined adsorption of these contaminants to plastics may rapidly elevate their toxicity."
Reviewer 2 Report
I have reviewed the manuscript "Microplastics in the Gulf of Mexico: A bird’s eye view", the study is very interesting, has a good design and structure. The authors clearly define their objective, hypothesis and methodology. Prior to further processing I request the authors to comply with the following comments, this will serve to improve their manuscript:
In general the literature review conducted is interesting and very well designed. The analysis of the current situation in the Gulf of Mexico and the effects make the reality of the object of study very clear.
In the last paragraph, make it clearer that a literature review was conducted, describe whether it was of scientific and/or gray literature.
I would ask the authors to explain a little more in one of the sections or to elaborate on the policies, laws or plans that address measures to prevent micro plastic problems.
The section on "conclusions and future directions" should be separated into challenges and then conclusions.
In reading about tourism activity in the northern gulf area, I would like the authors to address a bit about tourism activity and its influence on the presence of debris (only if possible).
Considering that microplastics is the final curve of the disintegration of macroplastics, I recommend reading this document will help them to address measures and recommendations. It could also be useful for the introductory section, first paragraph.
https://doi.org/10.1016/j.marpolbul.2019.02.003
Overall a good job
Author Response
We thank the reviewer for their comments on our manuscript and provide a detailed response to each comment, below.
In general the literature review conducted is interesting and very well designed. The analysis of the current situation in the Gulf of Mexico and the effects make the reality of the object of study very clear.
- Thank you!
In the last paragraph, make it clearer that a literature review was conducted, describe whether it was of scientific and/or gray literature.
- We have added a statement to this effect in the last paragraph of the introduction (lines 91-96): “Here, we review existing scientific peer-reviewed and gray literature (e.g., theses, dissertations, governmental and non-governmental reports) to provide a brief overview of the state of knowledge of microplastic pollution in the GoM, followed by a summary of known effects of microplastics and associated chemicals on birds, and methodological considerations for field and laboratory studies, including priority bird species in the GoM for field studies.”
I would ask the authors to explain a little more in one of the sections or to elaborate on the policies, laws or plans that address measures to prevent micro plastic problems.
- We have added a new section to our conclusions titled “Policy and mitigation” and written a new paragraph in this section (lines 571-586) focused on initiatives for mitigation in the Gulf of Mexico, and policies regarding plastics in the northern GoM (which are primarily antagonistic toward mitigation)
The section on "conclusions and future directions" should be separated into challenges and then conclusions.
- We have subdivided this section into 3 sub-sections now named: "Policy and mitigation," "Current gaps in knowledge," and "Recommendations for future studies".
In reading about tourism activity in the northern gulf area, I would like the authors to address a bit about tourism activity and its influence on the presence of debris (only if possible).
- We have added a parenthetical comment in lines 108-109: "Moreover, the GoM is subject to additional plastic discharges from the continents, port areas, tourism activities (which can have both positive [3] and negative effects [16,17] on beach cleanliness), river systems, and industrial activities [12]"
- The influence of tourism on plastic debris is complex - with positive effects from enhanced waste management in some locations and negative effects from increased discharge in other locations. Unfortunately, studies in the GoM focused on the impact of tourism on marine debris are quite limited. Those that do exist point to an increased accumulation of plastic debris in areas of high tourism (probably due to poor waste management and increased single-use plastic consumption by tourists). Anecdotally, some counties in the GoM have implemented enhanced waste-management strategies to benefit tourism activities with great success. However, we feel that this is beyond the scope of this review, especially considering the little scientific data currently in existing regarding this issue in the GoM.
Considering that microplastics is the final curve of the disintegration of macroplastics, I recommend reading this document will help them to address measures and recommendations. It could also be useful for the introductory section, first paragraph. https://doi.org/10.1016/j.marpolbul.2019.02.003
- We thank the reviewer for this suggestion and have added this citation to the first paragraph (lines 33-35) "Disposed and discarded plastics at various stages of breakdown are contributing to the growing quantities of litter on beaches and coastal areas, particularly on continents."
- We have also added it to our Conclusions section (lines 565-568) " Tourism appears to provide a strong incentive for coastal waste management (e.g., [3]) and increased emphasis on tourism and wildlife activities may convince policymakers and local parties to implement clean-up and disposal activities. "
Round 2
Reviewer 2 Report
I am grateful for the receptiveness of the authors. They have complied with the recommendations provided. The manuscript can be published after editorial review.